# Coccidioidomycosis Granulomas Informed by Other Diseases: Advancements, Gaps, and Challenges

**DOI:** 10.3390/jof9060650

**Published:** 2023-06-09

**Authors:** Nadia Miranda, Katrina K. Hoyer

**Affiliations:** 1Quantitative Systems Biology Graduate Program, University of California Merced, Merced, CA 95343, USA; nmiranda8@ucmerced.edu; 2Department of Molecular and Cell Biology, School of Natural Sciences, University of California Merced, Merced, CA 95343, USA; 3Health Sciences Research Institute, University of California Merced, Merced, CA 95343, USA

**Keywords:** granuloma, Valley fever, *Coccidioides*, chronic granulomatous disease, tuberculosis, sarcoidosis, NADPH

## Abstract

Valley fever is a respiratory disease caused by a soil fungus, *Coccidioides*, that is inhaled upon soil disruption. One mechanism by which the host immune system attempts to control and eliminate *Coccidioides* is through granuloma formation. However, very little is known about granulomas during *Coccidioides* infection. Granulomas were first identified in tuberculosis (TB) lungs as early as 1679, and yet many gaps in our understanding of granuloma formation, maintenance, and regulation remain. Granulomas are best defined in TB, providing clues that may be leveraged to understand *Coccidioides* infections. Granulomas also form during several other infectious and spontaneous diseases including sarcoidosis, chronic granulomatous disease (CGD), and others. This review explores our current understanding of granulomas, as well as potential mechanisms, and applies this knowledge to unraveling coccidioidomycosis granulomas.

## 1. Introduction to Coccidioidomycosis

Valley fever, or coccidioidomycosis, is a respiratory disease caused by inhaling *Coccidioides immitis* or *Coccidioides posadasii*. Coccidioidomycosis, first discovered around 130 years ago, has remained in the top three infectious diseases in California and Arizona for the past 10 years and is categorized as an emerging disease [1,2]. *Coccidioides* is endemic to the western and southwestern US and is predicted to spread due to global warming [3]. Sixty percent of *Coccidioides* infections are mild to asymptomatic with flu-like symptoms that typically resolve without diagnosis. Forty percent of infections are diagnosed with mild to severe symptoms that can last for months to years [4]. Around 5–10% of symptomatic patients develop disseminated disease often requiring lifetime treatment. Granuloma cell masses are identified in X-rays and MRI scans only to often be misdiagnosed as tumors, requiring confirmation by tissue biopsy [5,6]. Computerized tomography (CT) imaging is used to identify and monitor granulomatous tissue. Granulomas are also analyzed using fungal culture, microscopy via hematoxylin and eosin (H&E), and periodic acid Schiff (PAS) staining. There are limited coccidioidomycosis granuloma studies, leaving many unanswered questions. With increasing patient numbers, new research is critically needed to understand the immunological responses in this disease.

*Coccidioides* is a dimorphic fungus that grows as mycelia in alkaline, dry soil with sparce rainfall throughout the year. When the soil becomes dry, mycelia segments into arthroconidia (2–5 µm in diameter). Upon soil disruption, arthroconidia becomes aerosolized. In the lungs, arthroconidia transitions into spherules that are poorly engulfed by macrophages, likely due to the large spherule size (100–200 µm diameter). At maturation, spherules rupture releasing 100–400 endospores continuing the fungal growth and spread within the body [7,8]. In this review, we evaluate the granuloma literature, drawing conclusions and hypotheses in the context of coccidioidomycosis.

## 2. Granuloma Classification

Granulomas are thought to form to control pathogens that cannot be cleared, or to prevent excessive immune damage to the host from ongoing inflammatory responses [9]. If a granuloma forms due to an infection, it is called a caseating granuloma. A spectrum of granulomas has been observed during infection from highly necrotic to fibrotic, likely due to the host microenvironment, host inflammatory response, and pressure by the pathogen such as chronic granuloma disease (CGD), TB, or *Coccidioides*, and non-caseating inflammatory disease such as sarcoidosis (Figure 1a). In general, the mantle (edges) of a granuloma is usually made up of adaptive cells, while the middle encompasses innate cells. Fibrosis is associated with excessive wound healing and scarring which are typically regulated by anti-inflammatory responses [9,10,11]. In contrast, necrotic granulomas are associated with inflammatory cytokine production [9,12]. It is hypothesized that necrotic granulomas occur due to tumor necrosis factor (TNF) dysregulation triggering macrophages to undergo programmed necrosis though mitochondrial reactive oxygen species (ROS) [13]. Necrotic granuloma formation is inversely associated with patient morbidity in TB. Other pathogenic fungi produce necrotizing granulomas such as *Cryptococcus* and *Histoplasma* [14]. Necrotic granulomas are found in *Coccidioides* infection, and in recent studies, neutrophils have been found to accumulate around the center of *Coccidioides* granulomas, which are called pyogranulomas [15,16].

One uniting feature of all granulomas is the presence of differentiated, often atypical, macrophages. Foamy cells are macrophages that accumulate cytoplasmic oxidized lipids or lipid droplets and usually accumulate around necrotic material within a granuloma [17]. Multinucleated giant cells (MGCs) or Langhans giant cells are usually three or more uniformly shaped nuclei in a cell bigger than a mononuclear leukocyte [9]. Macrophages that undergo epithelioid transformation in a granuloma are distinctive with a flattened appearance, diffused cytoplasmic staining, elongated nuclei, and E-cadherin expression forming junctions (Figure 1b) [9]. Interestingly, macrophages, but not neutrophils, from any tissue type can differentiate into foamy cells in vitro [18]. It is hypothesized that macrophage failure to undergo cytokinesis leads to replication stress and activates DNA damage promoting macrophage transformation. This is supported by DNA damage observed in TB granulomas, lymph nodes, and skin sarcoid granulomas [19,20]. Although macrophages constitute the initial mass of cell type found within granulomas, there are other cell types that make up the rest of the granuloma.

Non-traditional immune cells, such as fibroblasts and endothelial cells, contribute significantly to shaping immune reactions within granulomas [21]. The number of immune cell types within a granuloma defines the granuloma complexity (Figure 1c). It is unknown what regulates inflammatory or anti-inflammatory responses forming within granulomas or surrounding tissues. However, it is hypothesized that granuloma maintenance needs pro-inflammatory cells in the middle to control the pathogen, and anti-inflammatory cells on the outside to aid in wound repair. Scientists often use the Goldilocks fairytale analogy to describe granulomas mechanistically: too much, too little, and just the right amount of inflammation. A balance in inflammation is necessary to maintain pathogen encapsulation, prevent spread, and support wound repair and cell turnover replacement (Figure 1d). The local tissue microenvironment determines individual granuloma structure and composition, differing greatly between diseases, and within an individual. Granuloma maintenance mechanisms are mostly unknown, and what is known is defined primarily during TB. More data are essential for other diseases as TB infection does not fully model granulomatous diseases caused by other pathogens including *Coccidioides*.

## 3. Granulomatous Diseases

There are multiple inflammatory diseases which trigger spontaneous granulomas to form even in the absence of infection. This review concentrates on infection-induced granulomas and sarcoidosis, which spontaneously triggers granuloma formation. Chronic granulomatous disease (CGD) is caused by mutations in nicotinamide adenine dinucleotide phosphate (NADPH) and form infectious granulomas. The best-studied granuloma are in TB and is applied to model other granulomatous diseases. There is a need to study other diseases as not all granulomas are triggered by infections or form the same response to different pathogens. Evaluating granuloma literature from other diseases provides foundational theories for exploration to define *Coccidioides* granuloma formation and maintenance.

### 3.1. Tuberculosis

TB is a deadly bacterial infection caused by *Mycobacterium tuberculosis* that infects the lungs and can spread through the air when an infected person coughs or sneezes. New data from positron emission tomography (PET) scans and CT scans suggest TB propagation starts in the lung cavitation and seeds a new infection via bronchial spread driving host disease progression and transmission [22]. TB forms multiple granuloma types that are misdiagnosed as cancer in chest images requiring an invasive pulmonary biopsy, potentially delaying treatment (Figure 2) [22,23,24]. It is estimated that TB treatments have suffered as resources were reallocated towards the COVID pandemic and because symptoms between these diseases are similar causing misdiagnoses. TB is treated with antibiotics, and for drug-resistant infection, drug-resistant antibiotic treatments exist but with harsh side effects.

TB granulomas have diverse outcomes even within an individual (for more details on TB granulomas, see reviews by Elkington, P. et al. (2022) [22], Miranda, M. et al. (2012) [25], and Ndlovu, H. et al. (2016) [26]. Pro- and anti-inflammatory responses are needed to fight this bacterial infection and require a precise and delicate balance. In non-human primates, a single individual can have over ten granulomas, each dramatically different in morphology and cellular composition [21]. Therefore, it is hypothesized that the localized tissue microenvironment plays a role in granuloma outcomes. RNA-sequencing data suggest lower colony forming units (CFU) in granulomas equate to greater bacteria killing rather than reduced bacterial growth [21].

Tregs, T cell exhaustion, and high anti-inflammatory responses might have a negative role within granulomas. Granulomas that appear later in the infection disease progression (termed late appearing granulomas), compared to granulomas that develop earlier, form an effective adaptive response that recruits T cells and kills TB better even in low-dose macaque infections [21]. Higher neutrophil presence in granulomas correlates with increased pathogen killing and host survival in TB [9]. Unfortunately, even in the same individual, each TB granuloma can contain different states of T helper (Th 1) and Th17 responses in TB control [21]. A recent human TB granuloma study found the granuloma border to be regulated by PD-L1 expression and IDO1 secretion, a novel finding [27]. Programmed cell death protein 1 (PD-1)–programmed death-ligand 1 (PD-L1) interactions are inhibitory signals regulating immune responses, and in cancer, expression of PD-1/PD-L1 or indoleamine 2, 3-dioxygenase 1 (IDO1) allows for an escape from immune surveillance [28]. PD-L1 expression is found across sarcoidosis, TB, and *mycobacteria avium* granulomas; however, IDO1 and PD-L1 co-upregulation is TB granuloma specific, and PBMC expression correlates with active TB [27]. PD-L1 blockade exacerbates TB disease in patients [29,30]. Thus, IDO1 and PD-L1 may provide a biomarker of disease but not a treatment option in TB, to distinguish between those with advancing disease compared to those with latent infection [27]. During TB infection, T cell activation is decreased due to PD1 and lymphocyte-activation gene 3 (LAG3) expression; this is thought to control the immune inflammation to protect the host [27,31]. Expression of these markers within the granuloma may regulate the inflammatory response or provide an escape mechanism to the pathogen. This further fulfils the Goldilocks analogy of containing a balance of cell types in and around the granuloma. Granuloma imaging suggests local signals regulated by unique cellular infiltration within each granuloma determines morphology and structure.

Macrophages utilize induced nitric oxide synthesis (iNOS) to induce local inflammatory responses, and iNOS appears to regulate granulomas. TB granulomas upregulate iNOS, endothelial NOS (eNOS), and arginase 1 (Arg1) compared to non-granulomatous tissue [32]. Macrophages located outside the granuloma (near non-involved normal tissue) are less inflammatory than macrophages involved in the granuloma. iNOS and arginase co-expression is found throughout necrotic granulomas, and high Arg1 expression is found in the lymphatic cuff (T and B cell area around the edge of a necrotic granuloma). M2 macrophages express Arg1 and colocalize in necrotic granulomas, while M1 macrophages express iNOS and localize around the rim of non-necrotic granulomas [32]. M2 macrophages within the granuloma core likely control the infection locally, while the anti-inflammatory barrier surrounding the granuloma limits immune-mediated tissue damage and assists with wound repair in the tissue.

Overall, TB granulomas maintain a complex immune environment within and around the granuloma depending on bacterial load, within a wide range of disseminated and non-cavity diseases, and the unknown effective adaptive immune response to control TB [22]. This has provided clues to what could be happening with *Coccidioides* which also utilizes iNOS. However, iNOS deficiency has limited impact on *Coccidioides* fungal burden and host survival, but it may influence granuloma formation (see Section 3.5) [33,34]. From iNOS studies in TB granulomas, we infer that in coccidioidomycosis the arginase/NOS pathway might lead to competition in macrophage polarity influencing how granulomas develop in this infection. NOS expression downregulates T cell receptor (TCR) ζ expression; thus, iNOS expression in *Coccidioides* infection might reduce T cell activation, proliferation, and cytokine secretion, thereby regulating host adaptive responses [35].

### 3.2. Sarcoidosis

Sarcoidosis is a systemic immunological disease where granulomas spontaneously develop anywhere in the body and is diagnosed via biopsy, MRI, X-rays, and PET scans. Clinicians hypothesize that sarcoidosis is caused by dysregulation of antigenic response to an unknown antigen and may be due to genetics [36]. Common symptoms are dyspnea, cough, fatigue, cardiac and neurological symptoms, and specific symptoms in the organ containing the granulomas [36]. Sarcoidosis has two important time points, typically the first occurs at 25–40 years of age, with a second onset at 50+ occurring in mostly females [37]. Two-thirds of patients evolve through a self-reemitting disease within 12–36 months, while 10–30% of patients develop chronic disease needing prolonged treatment [37]. Current treatment focuses on targeting macrophages and T cells by inhibiting TNF-α, interleukin (IL)-1, interferon (IFN)-γ, and IL-6 [37,38].

Sarcoidosis granulomas (Figure 3) contain multinucleated giant cells, foamy macrophages, epithelioid macrophages, a rim of CD4 T cells, and sometimes CD8 T cells and B cells surrounding the granuloma rim [36,38,39,40]. Sarcoidosis granulomas form in four stages: initial accumulation, effector phase, resolution, or fibrosis progression [41]. Disease is characterized by elevated Th1 responses, exhausted immunity, and defective or possibly exhausted Tregs [36,42,43,44]. High IL-2 is released by activated T cells during active disease and low IL-2 in chronic sarcoidosis; therefore, IL-2 is an early marker in active sarcoidosis [41]. iNOS is expressed in alveolar macrophages in sarcoidosis granulomas, while eNOS is present in epithelial cells [41]. Although studies are limited, we see similarities between iNOS in spontaneous granuloma and TB-induced granulomas, providing clues for exploration in coccidioidomycosis. Since iNOS is found in all three granulomatous diseases (TB, sarcoidosis, *Coccidioides*), iNOS likely is an underlying, necessary mechanism for granuloma formation.

### 3.3. Chronic Granulomatous Disease (CGD)

CGD is caused by an inherited X-linked autosomal recessive mutation in one of five NADPH oxidase genes. NADPH oxidase is a multiunit phagosome and plasma membrane-associated enzyme utilized for phagocytosis, and each of the five subunits influences specific disease symptoms [41]. Disease can be identified within the first few months of life during early exposure to pathogens. Focusing on disease in the lung, general symptoms include an inability to clear infections that induce granulomas and autoimmune-like symptoms (Figure 3) [41]. Disease is confirmed by sequencing for NADPH oxidase mutations and via a dihydrorhodamine 123 (DHR) test (a white blood cell ROS indicator assay). Antibiotics, antifungals, IFNγ, or steroids are used to treat the infections that CGD patients develop. Although a complex disease, this review focuses on lung aspects of the disease to model coccidioidomycosis in the lung. For a detailed review on CGD, see [45].

CGD macrophages are largely pro-inflammatory. In zymosan-induced peritonitis, a CGD mouse model, macrophages cannot downregulate pro-inflammatory cytokines or upregulate oxidative phosphorylation, resulting in phenotypically and transcriptionally immature macrophages [46]. iNOS is an important effector molecule in macrophages and is defective in CGD, removing a critical mechanism of CGD macrophage function. NADPH oxidase 2 (NOX2) in alveolar macrophages maintains alveolar homeostasis as spontaneous changes in transcriptome and epigenome occur in the first few months of life [47]. NOX2 deficiency in alveolar macrophages increases pro-inflammatory cytokines, and the alveolar macrophages become permanently altered by epigenetic modifications, promoting a dysregulated trained immune response. CGD provides hints to possible macrophage mechanisms in granulomas via the NADPH oxidase pathway [48]. A recent *Coccidioides* patient study identified polymorphisms in *DUOX1* or *DUOXA1* (NADPH subunits) polymorphisms as risk factors for severe disease [49]. Therefore, impaired fungal recognition or cellular responses to *Coccidioides* are correlated with disseminated disease [49,50].

### 3.4. Coccidioidomycosis Immunological Responses

Several innate and adaptive immune populations respond to *Coccidioides* infection (see Diep et al. (2020) [51] for detailed review). Polymorphonuclear neutrophils (PMNs) are the first immune responders to migrate into sites of infection, although tissue-resident immune cells initiate immunity [52,53,54,55]. Based on in vitro studies, human PMNs have a limited ability to phagocytose and kill arthroconidia, spherules, and endospores, although the rate of phagocytosis depends on the *Coccidioides* strain [56]. In vitro, human neutrophils phagocytose endospores with partial phagocytosis of larger spherules [57]. Dendritic cells (DCs) survey the environment for pathogens, then activate T cells via antigen presentation and co-stimulation. DCs are polarized towards a DC1 state during in vitro *Coccidioides* interactions but have reduced activation [58]. In TB, DCs transition into DC1 and continuously migrate within and out of granulomas providing antigen presentation to control local T cell responses [59]. Mice treated with dexamethasone (induces an immunocompromised state) and infected with *Coccidioides* had reduced immune cell numbers (T cells, B cells, macrophages, DCs, and neutrophils), compared to untreated mice [16]. Vaccination with the attenuated *Coccidioides* strain (Δcps1) provides protection against virulent *Coccidioides* challenge that is neutrophil dependent through the inhibition of endospore and spherule growth via oxidative burst in mice [60]. In vaccinated mice challenged with virulent *Coccidioides*, neutrophil and eosinophil lung infiltration is increased compared to non-vaccinated mice. Vaccinated mice also exhibit increased lung-infiltrating macrophages and dendritic cells [61]. Mouse vaccine studies have not focused on lung granuloma formation, although dog vaccination shows no evidence of granulomas. Vaccination in dogs shows no evidence of granuloma, irritation, or side effects. Dogs that receive vaccination and boost, however, largely develop pyogranulomatous lesions in their lungs and lymph nodes upon *Coccidioides* infection [16].

Innate cells detect *Coccidioides* via pathogen recognition receptors (PPR), specifically through Dectin-1 and TLR2 (Figure 4) [8,34,62,63,64,65,66,67,68,69,70,71,72,73,74]. β-1-3-glucan, chitinases, and SOWgp make up the *Coccidioides* cell wall in early spherule stages which are detected by PRRs [75]. On innate immune cells, the Dectin-1 pathway continues through MyD88 and CARD9, leading to the activation of transcription factors such as NFκB, NFAT, and IRF4 [76]. Pro-inflammatory cytokines produced from macrophages in this pathway include MIP2, CXCL2, GM-CSF, TNF-α, IFN-γ, and IL-23 [77]. NFκB also activates iNOS, leading to nitric oxide (NO) production (Figure 5) [69,71]. Macrophages phagocytose arthroconidia [78,79] and are hypothesized to induce *Coccidioides* transition into spherules. Once engulfed, macrophages generate ROS and NOS to degrade *Coccidioides* [65]. However, macrophages can only phagocytose up to ~20 µm in diameter, and as spherules mature, they become too large for engulfment (100–200 µm). In vitro, macrophages phagocytose *Coccidioides*, but are blocked at the M0 differentiation phase [58,80,81]. Additional research in *Coccidioides* mechanisms is needed to understand how *Coccidioides* manipulates macrophages.

*Coccidioides* adapts to its environment by switching into its parasitic life cycle within the lungs. *Coccidioides* evades PRR detection by degrading its cell wall component SOWgp. SOWgp is degraded via MEP1 production from *Coccidioides* during the spherule state to release new endospores [73]. It is hypothesized that *Coccidioides* survives in mammals through iNOS production blockade by fungal secreted urase, causing the alkalization of the environment, and forcing competition on L-arginase between urease and iNOS [82]. L-arginase skewing towards arginase leads to more urase production and anti-inflammatory cytokines such as IL-10, IL-4, and IL-13. High anti-inflammatory cytokines polarize macrophages into an anti-inflammatory state (M2) (Figure 4). *Coccidioides* seems to manipulate the immune system, specifically macrophages, to survive. Monocytes become M0 but do not polarize into M1 nor M2 subtypes in the presence of *Coccidioides* in vitro. It is possible that *Coccidioides* blocks macrophage phagocytosis and polarization to prevent cytokine release to evade innate immunity [58]. In vitro stimulated DCs and macrophages cultured with *Coccidioides* express less MHC-II/CD86 co-expression, suggesting *Coccidioides* is directly inhibiting activation and maturation; fungi are known to manipulate macrophage polarization to increase survival in their host [83,84]. *Coccidioides* can thrive within necrotizing granulomas perhaps by feeding off of dead material as suggested in decaying mammals within the soil [85]. 

DCs and innate immune cytokines shape the development T helper response to facilitate a functional antigen-specific protection. T cells elicit Th1 and Th17 responses towards *Coccidioides* [61,86], producing IFN-γ and TNF-α (Th1), and IL-17 (Th17) [87,88]. Anti-inflammatory IL-10 production is likely produced by Th2, Tregs, and macrophages, while pro-inflammatory IFN-γ may derive from Th1 and natural killer cells. Th1 and Th17 responses are effective at controlling *Coccidioides* infection, except in situations of immunosuppression or immune modulating signaling defects. Vaccine studies in mice produce effective *Coccidioides* immunity. Further research is needed to identify the mechanisms *Coccidioides* utilizes to survive within the host and evade host immunity.

### 3.5. Lung Granulomas in Coccidioidomycosis 

There are a limited number of *Coccidioides* granulomas studies, many of which are from the 1960s and are not easily accessible online. Granulomas form to control larger spherules that cannot be phagocytosed. Complex, necrotizing granulomas made up of macrophages, neutrophils, and T and B cells often form in response to *Coccidioides* [89]. *Coccidioides* mouse granulomas include neutrophils [89] and contain a necrotizing core with dead/dying cells (Figure 5) [9]. *Coccidioides* endospores and spherules are encapsulated within granulomas. It is unknown if mycelia can grow within the granuloma center as mycelia have been found in pulmonary cavities [90]. Mouse neutrophils in the skin are associated with spherules [91], although additional quantification and studies are needed to define these interactions. CD14+ macrophages localize primarily inside the granuloma rather than the mantle or in the surrounding tissue, while CD206+ macrophages are equivalent within and around granulomas [33]. However, CD14, CD206, IL-10, and TNF-α expression is higher within the granuloma compared to the surrounding tissue [33]. More studies are needed to solidify these findings and compare skin and lung granulomas.

Patient samples have revealed that CD4 T cells, CD20 B cells, and IFN-γ are present in the mantle regions of necrotizing *Coccidioides* granulomas [89]. Human granulomas contain CD3 lymphocytes that outnumber B cells in the mantle region. IFNγ (from Th1 and NK cells) expression is higher within the mantle, while CD4 T cells and B cells associate with IL-10 production [33,89]. Th2 responses localize in the mantle, while Th1 responses localize in the middle of the granuloma. However, this Th1 and Th2 response in *Coccidioides* granulomas are the opposite of that seen in TB. In an acute coccidioidomycosis study, 87% of patients had ill-defined granuloma shapes, while 13% of nodules (granulomatous tissue) were smooth [92]. Well-defined and coalescence of nodules was found in 6 out of 15 patients [92]. In chronic infection, multiple lung nodules can be observed via CT images [33]. Disseminated patient skin biopsies also contained CD4 T cells in the middle and CD8 T cells in the mantle of granulomas [93]. Organized *Coccidioides* granulomas likely provide pathogen containment via Th1 cytokines and protective wound healing responses from excessive inflammation via Th2 cytokine responses.

Induced immune suppression via dexamethasone reactivates *Coccidioides* via loss of granuloma formation, highlighting the importance of immune cells in maintaining granuloma formation and ongoing interactions for fungal control. This study begins to unveil the control of lung granuloma maintenance during *Coccidioides* infection by the immune system [16]. The mantle of well-regulated granulomas contains macrophages, fibroblasts, and T and B cells. Neutrophils are located in pockets around the center, forming a pyogranuloma [16]. *Coccidioides* can be found in each life cycle stage within granulomas, although the parasitic life cycle (spherules and endospores) is by far the most common and easiest to diagnose [8]. In chronic infection, *Coccidioides* can establish favorable conditions in the lung such as arginase activity, alkaline pH (urase), lower O_2_ concentrations, tissue damage, and immune system evasion with fungal morphological changes (accompanied by surface glycogen changes) that support growth and disease progression (Figure 4) [8]. Further granuloma investigation is needed to define important cell types and phenotypes, to understand what promotes the fungal and immune signals that support and allow chronic infection or dissemination in coccidioidomycosis. 

iNOS is possibly involved in macrophage phagocytosis mechanism in coccidioidomycosis. High IL-10 production in the lungs impairs resistance to *Coccidioides*. IL-10-deficient mice produce increased iNOS and clear *Coccidioides* more effectively than control mice [34]. During *Coccidioides* clearance, iNOS-deficient mice have no differences in IL-10, IL-6, IL-17, and GM-CSF on day 7 and 11 relative to control mice [70]. Vaccinated control and iNOS knock out mice maintain similar survival and fungal burden until day 11, when iNOS mice begin to struggle with containing the fungal burden [70]. The lack of iNOS likely results in an inability to effectively phagocytose *Coccidioides* properly or form and maintain effective granulomas. iNOS may not increase host survival, but instead helps maintain granulomas and inhibit dissemination. Based on TB and granulomatous disease studies, *Coccidioides* likely inhibits the NADPH oxidase pathway to contain virulence within the lungs. NADPH oxidase also seems to play a role in *Coccidioides* granuloma formation as described above in CGD, sarcoidosis, and TB.

## 4. Limitations, Challenges, and Gaps in Granulomas

Granulomas vary in size, architecture, and immune composition. Many unknowns remain regarding granulomas, and many limitations and challenges slow our understanding. Granulomas form within deep lung tissue requiring biopsies to analyze. Traditionally, H&E and PAS staining is performed to identify basic cell types and pathogens within granulomas. TB granuloma studies performed in non-human primates have refined our understanding of how granulomas are formed and maintained [21]. Performing these studies with *Coccidioides* would be beneficial for identifying which immune cells interact with *Coccidioides* in the lungs (and other tissues during dissemination), and how immune cells function to control infection. Skin and bone lesions occur in disseminated coccidioidomycosis. The underlying mechanisms and etiology are unknown. Also missing is an analysis of similarities and differences between lung, skin, and bone granulomas and comparisons of granulomas between humans, dogs, and mouse infections.

Technology has advanced with new techniques that offer more specificity to better understand these diseases. Flow cytometry offers options to specify cell subsets; however, granulomas contain multinucleated macrophages and would typically be eliminated from analysis as cell doublets due to their large size. Multinucleated macrophages are known to be very sticky, and possibly more sensitive to apoptosis. Therefore, multinucleated macrophages might be sensitive to digestion. *Coccidioides* granulomas are typically necrotizing and might skew data by over-representing dead cells. Advancements in flow cytometry such as imaging flow cytometry could aid in observing multinucleated cells and for better identification of cellular subsets.

Other technological advancements such as spatial sequencing will carry out huge benefits for coccidioidomycosis studies. Spatial sequencing offers RNA-sequencing with tissues such as granulomas and provides cell identification, gene expression per cell, and localization relative to the pathogen. Thus, spatial sequencing provides a unique perspective into the tissue environment within the granuloma [94]. Flow cytometry could be combined with spatial sequencing, providing localization and detailed cell identification. Another newer technology that could be used is tissue clearing [95]. Whichever new technique is used, it should involve the extracellular matrix, fibroblasts, endothelial cells, and other cells that are not typically considered immune cells to understand their involvement in granuloma formation and maintenance. Future studies need to utilize larger sample sizes or modeling to increase statistical analysis, as most published studies utilize a small sample size due to the rarity of disease and sample access.

## 5. Future Directions for Coccidioidomycosis and Conclusions

Granulomas likely form due to the inability of the immune system to clear a pathogen or to protect the host from immune-mediated damage that would be required for pathogen clearance. They are complex structures that contain different types of macrophage transformations; the more different cell types, the more complex the granuloma. It has become clear in recent studies that granuloma maintenance is supported by both anti- and pro-inflammatory cells and cytokines to aid in pathogen clearance within the granuloma and in wound repair outside the granuloma. Therefore, complex microenvironments greatly impact the future of each individual granuloma. Mechanisms revolving around how granulomas form and are maintained are largely unknown. TB is an ancient and ongoing disease and sets the baseline for other granulomatous diseases. More research is needed to understand the other granulomatous diseases as they are not all initiated by the same pathogen. Some granulomas even spontaneously occur, likely using both similar and different mechanisms from infection. Sarcoidosis is caused by spontaneous granuloma formation and is poorly understood, leading to limited treatment options. On the other hand, CGD is caused by an X-linked mutation on any NADPH oxidase subunit that causes granuloma formation when a pathogen is present. CGD likely causes an unregulated pro-inflammatory microenvironment, leading to dysregulated trained immunity.

Currently, no fungal vaccines exist, although a dog vaccine is being evaluated for coccidioidomycosis, and many research groups are focused on vaccine development for *Coccidioides* in humans [96]. Further research is needed to understand the effective immunological responses towards *Coccidioides* and to determine the level of protection during re-exposure. Recent research indicates a correlation between impaired pathogen recognition and responses to chronic or disseminated disease outcomes. Patients with recognition and response complications are likely due to genetic mutations in PRR pathways [49]. Tying back to the other granulomatous diseases mentioned throughout this review, all associate with the NADPH oxidase pathway. The NADPH oxidase pathway regulates phagocytosis and metabolomics within macrophages, and studies seem to indicate an important mechanism involved in granuloma maintenance. Further research is needed to explore the NADPH oxidase pathway in coccidioidomycosis granulomas and identify other mechanisms and cell types crucial in controlling *Coccidioides* infection within granulomas. 

## Figures and Tables

**Figure 1 jof-09-00650-f001:**
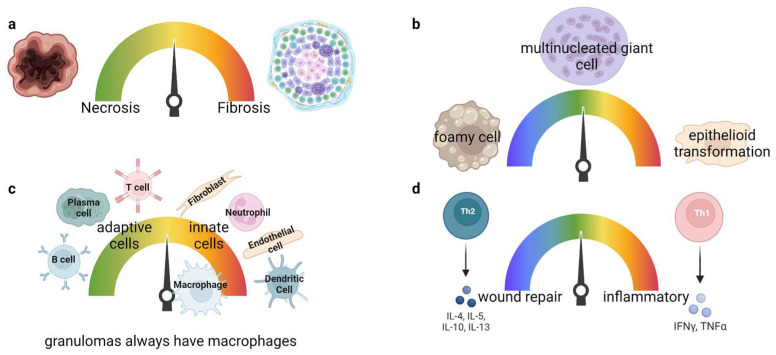
Granuloma classification and architecture. Granulomas form in a spectrum where (**b**–**d**) can be found within a single granuloma. (**a**) A mature granuloma can either be necrotic or fibrotic. Necrotic granulomas encompass a core of many dead cells alongside the pathogen. Fibrotic granulomas are well structured, maintained, and contain extracellular matrix surrounding the granuloma. (**b**) Macrophage transformations. Foamy cells contain lipid droplets in the cytoplasm. Multinucleated macrophages are multiple macrophages fused together, hence containing multiple nuclei. Macrophages that undergo epithelioid transformation are elongated cells resembling epithelioid cells that form junctions. (**c**) Macrophages are the major cell types forming a granuloma. Traditional and non-traditional immune cells constitute the granuloma, and the number of immune cells defines the granuloma complexity. (**d**) Granuloma immune cells can produce pro- and anti-inflammatory responses to maintain the pathogen encapsulated in the middle and aid in wound repair in the surrounding mantle.

**Figure 2 jof-09-00650-f002:**
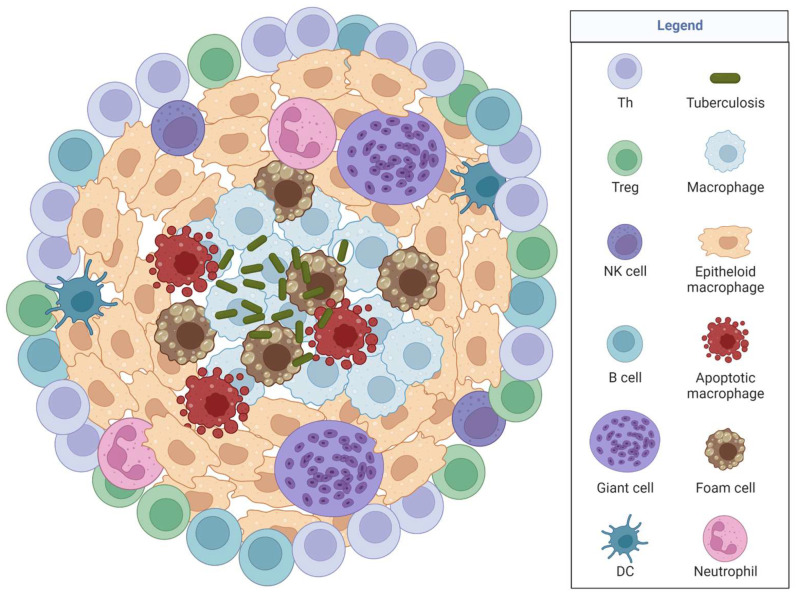
TB granulomas include *Mycobacterium tuberculosis* encapsulated in the middle surrounded by different cell types, always including macrophages, and sometimes including other innate and adaptive cells depending on granuloma progression.

**Figure 3 jof-09-00650-f003:**
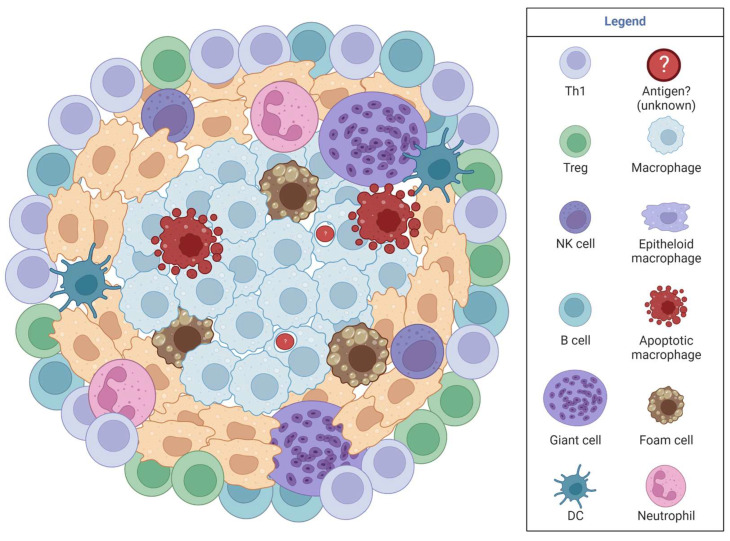
A typical sarcoidosis or chronic granulomatous disease granuloma (CGD). It is unknown if there is an antigen that impacts granuloma formation or spontaneous inflammation. Granulomas are typically sterile in the absence of an infection. CGD is caused by genetic defects in NADPH. Macrophages are the primary immune cells within a granuloma, with various other immune cells found to varying degrees within a granuloma, with unknown purpose.

**Figure 4 jof-09-00650-f004:**
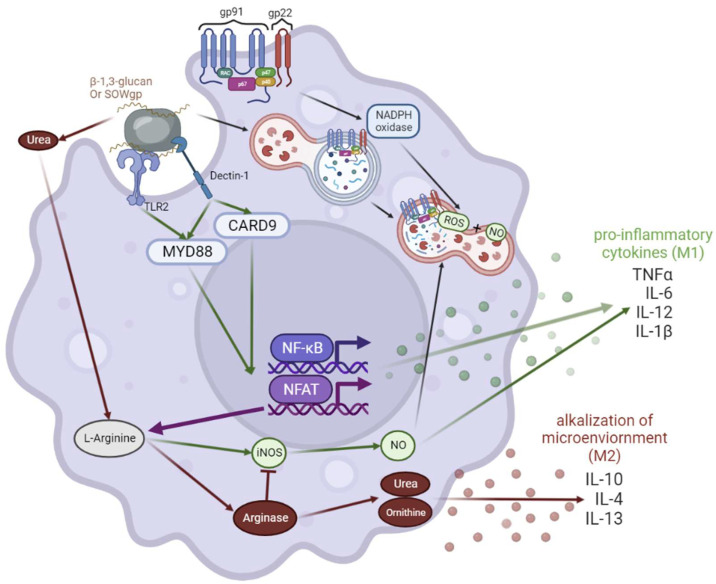
Macrophage phagocytosing *Coccidioides*. *Coccidioides* expresses SOWgp and β-1,3 glucan on its cell wall that can be detected via Dectin-1, a C type lectin receptor on innate cells. Dectin-1 induces signaling through MyD88 and CARD9, activating NFAT and NF-кB, transcription factors, that regulate pro-inflammatory cytokine production. On the other hand, *Coccidioides* releases urea within the microenvironment to promote survival within the host. Urea competes for L-Arginine against iNOS, leading to an alkalized environment and anti-inflammatory response. If a phagocytosing cell, such as a macrophage, is successful in capturing *Coccidioides*, it uses NADPH oxidases and ROS to promote *Coccidioides* degradation.

**Figure 5 jof-09-00650-f005:**
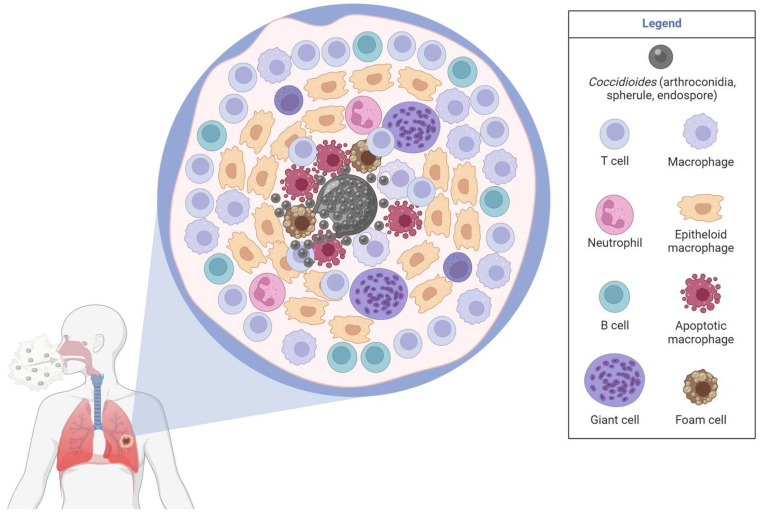
Coccidioidomycosis granuloma structure inferred by current literature and speculation from other granulomatous diseases. Granulomas develop in ~5% of *Coccidioides* infections and are mostly necrotic with encapsulated spherules or endospores that are walled off from the rest of the tissue by giant cells and other immune cells. Immune cells depicted (see legend) are thought to aid in fungal control (within the granuloma core) and prevent ongoing inflammatory responses and aid in wound repair (granuloma mantle). It is unknown if other immune cells (such as NK, DCs, or Tregs) are found in *Coccidioides* granulomas as in TB.

## Data Availability

Not applicable.

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
