# Peer review of "Coccidioidomycosis Granulomas Informed by Other Diseases: Advancements, Gaps, and Challenges"

_jof, 2023, doi:10.3390/jof9060650_

Round 1

Reviewer 1 Report

 This manuscript is an ambitious review of granuloma science and its application to the understanding of Coccidioides granulomas.  The topic is well-researched for the most part, but the organization of the material into the manuscript makes it difficult to understand the story of the granuloma, their structure, composition, and function, and how this leads the authors to the discussion of Coccidioides granulomas in particular.  I suggest reconsidering the organization of the entire manuscript and possibly being more selective of what is included in this manuscript.  For instance, the entire section on coccidioidomycosis of dogs does not fit into this manuscript outside of a single manuscript that addresses composition of canine granulomas. Also, I strongly urge the authors to drop the use of "Valley Fever", except to mention it once as the common name of this disease, and refer to it as coccidioidomycosis throughout the manuscript.  

Line 14 – punctuation problem.  After 1679, should be a comma, not a semicolon

Line 24 – either capitalize Valley and Fever if you want to refer to this as a proper name, or lowercase them both, and either would be acceptable.  There is no logic in the English language rules to have one of the start letters upper and one lower case unless you think it would also be correct to write the author’s name Nadia miranda.  Also, whatever correction you decide upon, please correct throughout the manuscript.

            Last, you could largely avoid the problem because it is most correct to refer to the disease as coccidioidomycosis in a technical paper such as this is.  That would be my primary suggestion.  Including reconsider the title.

Line 31 “60%” should be “Sixty percent” because it is the beginning of a sentence.  Same on line 32

Introduction is a bit disorganized.  The first paragraph skips rapidly from one epidemiological or clinical topic to the next, then switches back to describing how people get infected in paragraph 2, and then moves to a discussion of host immune responses.

Line 45 – What attenuated strain?  What kind of “protection”?  AIDS patients have no issues with neutrophils but die of fulminant or disseminated infection because control of this disease is largely adaptive and neutrophils do not provide protection.

Line 48, engulfed is spelled wrong.

Line 57 – this is two sentences with no punctuation in between.

Lines 60-61 – consider reread your reference 21 as they did excellent double staining that suggests a well-organized Th1 to Th2 balanced response from the center to the periphery of the human lung granuloma, and they propose that the Th1 centrally is controlling fungal proliferation while the Th2 on the margins may be preventing robust inflammation that is harmful to the host.  This older paper, with fewer reagents available, also suggests something of this nature in organization: In situ localization of T lymphocytes in disseminated coccidioidomycosis

R L Modlin, G P Segal, F M Hofman, M S Walley, R H Johnson, C R Taylor, T H Rea

Lines 66-76 don’t say anything important that drives this review.

Line 77.  You haven’t described the different kinds of granulomas or defined what a  caseating granuloma is.

Line 94 – monocular?

Line 101-102 does not have anything to do with the topic in line 103.  Why does this paragraph start with fibrosis and then not address it?

Line 104-105.  This sentence does not make sense in its current form.  Try rewriting it to help reader understand what you mean.

Figure 1: It doesn’t make sense to say that each granuloma type can be found within a single granuloma.

Section 3 – maybe the authors should start the review here, since you are mainly discussing a lot of section 2 in a vacuum and might be better able to talk about types of granulomas if you define the diseases and characteristics first.  If TB is going to be the defining granulomatous infectious disease because it has had the most defining research with most modern tools, then you need to spend more time on it early, then contrast and compare what is and is not known about cocci granulomas.

Line 260 – coccidioidomycosis is spelled wrong, and I don’t know why at this very late point in the manuscript the authors label this section with 3 names separated by a slash mark.  Decide what to call it that is consistent with the rest of the manuscript.

Line 262 – mantle, not mantel

Figure 4 is very nice except technically there are no Coccidioides spores in the center of a granuloma; there are spherules and endospores.

Line 292 – Dogs in Arizona are calculated to have a 28% likelihood of having been infected by Coccidioides by 2 years of age.  The incidence of infection was not compared to any other endemic or nonendemic regions.

Line 330 – this redundant, the authors already said this in the introduction.

Line 350 – urease? Also punctuation error again, get rid of semicolon and replace with comma

Section 4.1  I do not see that the discussion of canine valley fever adds anything at all to a paper on coccidioidal granulomas.  The single reference this reviewer was able to find on the study of dog granulomas does not suggest they are different than granulomas in other species and the content can be handled in the discussion of granulomas in humans and mice.

Line 379  - granulomas of TB (some) and leprosy form on the skin and are really easy to access, making study of these easy and one of the leading sources of modern information on granulomas.  See literature on lepromatous, RR and tuberculoid leprosy.  Also, this reviewer is not certain whether there is literature on granulomas of cocci in NHPs, but granulomas of NHPs infected with TB have provided a great deal of insight into mechanisms of pathogen control recently (Gideon HP, et.al, 2021 Multimodal profiling of lung granulomas reveals cellular correlates of tuberculosis control)

Line 421 – this is not a complete sentence. Neither is 429.  Although subordinates a clause, it is not the start of a new sentence in formal written English.

Author Response

Response to Reviewer 1 Comments

This manuscript is an ambitious review of granuloma science and its application to the understanding of Coccidioides granulomas.  The topic is well-researched for the most part, but the organization of the material into the manuscript makes it difficult to understand the story of the granuloma, their structure, composition, and function, and how this leads the authors to the discussion of Coccidioides granulomas in particular.  I suggest reconsidering the organization of the entire manuscript and possibly being more selective of what is included in this manuscript.  For instance, the entire section on coccidioidomycosis of dogs does not fit into this manuscript outside of a single manuscript that addresses composition of canine granulomas. Also, I strongly urge the authors to drop the use of "Valley Fever", except to mention it once as the common name of this disease, and refer to it as coccidioidomycosis throughout the manuscript. 

We thank the reviewer for their detailed review and suggestions. We have made major changes to the organization of our review and incorporated all suggestions.

Line 14 – punctuation problem.  After 1679, should be a comma, not a semicolon

Punctuation has been corrected.

Line 24 – either capitalize Valley and Fever if you want to refer to this as a proper name, or lowercase them both, and either would be acceptable.  There is no logic in the English language rules to have one of the start letters upper and one lower case unless you think it would also be correct to write the author’s name Nadia miranda. Also, whatever correction you decide upon, please correct throughout the manuscript.

Last, you could largely avoid the problem because it is most correct to refer to the disease as coccidioidomycosis in a technical paper such as this is.  That would be my primary suggestion.  Including reconsider the title.

‘Valley fever’ is now only used in the initial disease description. All other uses of Valley fever have been changed to coccidioidomycosis. Only Valley is capitalized in Valley fever as Valley refers to the San Joaquin Valley and fever refers to one of the major symptoms. This is the standard spelling and capitalization used in the field. See: “Valley fever.” Merriam-Webster.com Dictionary, Merriam-Webster, https://www.merriam-webster.com/dictionary/valley%20fever. Accessed 17 Mar. 2023.

Line 31 “60%” should be “Sixty percent” because it is the beginning of a sentence.  Same on line 32

We have made these corrections (still line 29).

Introduction is a bit disorganized.  The first paragraph skips rapidly from one epidemiological or clinical topic to the next, then switches back to describing how people get infected in paragraph 2, and then moves to a discussion of host immune responses.

Thank you for this recommendation. We have grouped and edited these topics in the introduction.

Line 45 – What attenuated strain?  What kind of “protection”?  AIDS patients have no issues with neutrophils but die of fulminant or disseminated infection because control of this disease is largely adaptive and neutrophils do not provide protection.

In this study, vaccinated mice who underwent neutrophil depletion showed protection via a neutrophil dependent oxidative burst in response to Coccidioidies challenge. We have clarified these details within the text (line 286).

Line 48, engulfed is spelled wrong.

Spelling errors have been fixed (now line 294).

Line 57 – this is two sentences with no punctuation in between.

Thank you, we have made this correction.

Lines 60-61 – consider reread your reference 21 as they did excellent double staining that suggests a well-organized Th1 to Th2 balanced response from the center to the periphery of the human lung granuloma, and they propose that the Th1 centrally is controlling fungal proliferation while the Th2 on the margins may be preventing robust inflammation that is harmful to the host.  This older paper, with fewer reagents available, also suggests something of this nature in organization: In situ localization of T lymphocytes in disseminated coccidioidomycosis R L Modlin, G P Segal, F M Hofman, M S Walley, R H Johnson, C R Taylor, T H Rea

(Now line 355): We have expanded our analysis of publication 21 (now 91) (now located in section 3.5, lines 339, 340, 351, and 354). Unfortunately, IL-10 is not considered soley a Th2 cytokine in humans, and can be expressed by Th1, Th2 and Tregs. The Li manuscript clearly shows that there are few CD25+Tregs in the human granuloma tissue tested, but other Bregs and CD8 Tregs may be present that have not been evaluated, and the Il-10 may also be secreted by the numerous Th1 (IFNg+ T cells) and macrophages. Thank you for suggesting the Modlin et al. paper, we incorporated their findings into section 3.5. Although this study looks at CD4 and CD8 T cells within granulomas, the authors were unable to determine cell function. Therfore, we do not know, in this specific study, what these T cells are expressing, whether a Th1 or Th2 effecgtor response. Although, IFNy is expressed in the mantle region of the skin granulomas.

Lines 66-76 don’t say anything important that drives this review.

This paragraph has been rearranged and incorporated into other section 2 paragraphs as some of the topics are important for understanding granuloma maintenence.

Line 77.  You haven’t described the different kinds of granulomas or defined what a caseating granuloma is.

We have clafrified the descriptions (now line 51) and described similarities and differences between granuloma types.

Line 94 – monocular?

Fixed misspelling; thank you for pointing out these errors as they identified an issue with our Word spell check.

Line 101-102 does not have anything to do with the topic in line 103.  Why does this paragraph start with fibrosis and then not address it?

A transition sentence was added to the pragraph to make transition between paragaphs smoother. This paragraph focuses on fibroblasts (now line 109) and describes these cells within granulomas (figure 1C).

Line 104-105.  This sentence does not make sense in its current form.  Try rewriting it to help reader understand what you mean.

Thank you we have clarified this section.

Figure 1: It doesn’t make sense to say that each granuloma type can be found within a single granuloma.

We have clarified the description of this figure.

Section 3 – maybe the authors should start the review here, since you are mainly discussing a lot of section 2 in a vacuum and might be better able to talk about types of granulomas if you define the diseases and characteristics first.  If TB is going to be the defining granulomatous infectious disease because it has had the most defining research with most modern tools, then you need to spend more time on it early, then contrast and compare what is and is not known about cocci granulomas.

Thank you for the suggestion. We prefer to describe overall granuloma structures and heterogenity before using the diseases to add layers to our understanding of these structures. Our experience is that this order is easiest for new-comers to this field to understand.

There are many well written TB granuloma papers, we have included our favorite reviews as references for the reader, however, we have limited space and have attempted to compare and contrast to TB within the limits of this review.

Line 260 – coccidioidomycosis is spelled wrong, and I don’t know why at this very late point in the manuscript the authors label this section with 3 names separated by a slash mark.  Decide what to call it that is consistent with the rest of the manuscript.

We have verified the spelling ot coccidioidomycosis throughout review.

Line 262 – mantle, not mantel

Fixed.

Figure 4 is very nice except technically there are no Coccidioides spores in the center of a granuloma; there are spherules and endospores.

This has been fixed.

Line 292 – Dogs in Arizona are calculated to have a 28% likelihood of having been infected by Coccidioides by 2 years of age.  The incidence of infection was not compared to any other endemic or nonendemic regions.

We have deleted the section focused on dogs.

Line 330 – this redundant, the authors already said this in the introduction.

Deleted this paragraph.

Line 350 – urease? Also punctuation error again, get rid of semicolon and replace with comma

Fixed.

Section 4.1  I do not see that the discussion of canine valley fever adds anything at all to a paper on coccidioidal granulomas.  The single reference this reviewer was able to find on the study of dog granulomas does not suggest they are different than granulomas in other species and the content can be handled in the discussion of granulomas in humans and mice.

Section deleted.

Line 379  - granulomas of TB (some) and leprosy form on the skin and are really easy to access, making study of these easy and one of the leading sources of modern information on granulomas.  See literature on lepromatous, RR and tuberculoid leprosy.  Also, this reviewer is not certain whether there is literature on granulomas of cocci in NHPs, but granulomas of NHPs infected with TB have provided a great deal of insight into mechanisms of pathogen control recently (Gideon HP, et.al, 2021 Multimodal profiling of lung granulomas reveals cellular correlates of tuberculosis control)

This review focuses on granulomas within the lung; similarities and differences between lung and skin granulomas in Coccidioides are unknown and not addressed in this review. There are unfortunately no published non-human primate sudies focused on Coccidioides granulomas. Although, there are publications of natural Coccidioides infection in NHP that indicate granulomas form, without firther exploration of the granuloma structure and regulation (see citations below). TB is more advances in this area, therefore we utilize TB to infer about Coccidoides granulomas. The Gideon paper is an amazing advancement to the granuloma field and we described this work section 3.1 (was citation 33, now 21).

Kundu, M. C., Ringenberg, M. A., d'Epagnier, D. L., Haag, H. L., & Maguire, S. (2017). Coccidioidomycosis in an Indoor-housed Rhesus Macaque (Macaca mulatta). Comparative medicine, 67(5), 452–455.

Koistinen, K., Mullaney, L., Bell, T., Zaki, S., Nalca, A., Frick, O., Livingston, V., Robinson, C. G., Estep, J. S., Batey, K. L., Dick, E. J., Jr, & Owston, M. A. (2018). Coccidioidomycosis in Nonhuman Primates: Pathologic and Clinical Findings. Veterinary pathology, 55(6), 905–915. https://doi.org/10.1177/0300985818787306

Line 421 – this is not a complete sentence. Neither is 429.  Although subordinates a clause, it is not the start of a new sentence in formal written English.

Fixed these two sentences.

Reviewer 2 Report

General comment

The manuscript "Valley fever and other infectious granulomas: Advancements, gaps, and challenges" describes the current knowledge of granulomas in a general context, and particularly in Valley fever. A topic of great relevance, fundamentally for the medical area; however, I have several comments:

1) I suggest changing the title since the way it is written suggests that the central approach is granulomas in Valley fever; however, although I know that the information is very scarce on granulomas-Valley fever, most of the manuscript is focused on granulomas associated with other pathologies, which is where this clinical aspect is described in greater detail.

I suggest the title "Hypothesis on the presence of granulomas in Valley fever concerning other infectious and spontaneous diseases: Advances, gaps, and challenges."

2) In section 3 “Granulomatous diseases,” it should be included as a subsection “3.4 Coccidioidomycosis.”

3) In the section “4. Coccidiomycosis/Valley fever/Coccidioides” lines 280-282, the authors mention that more research on granuloma is needed to define essential cell types and phenotypes, to understand how chronic infection or dissemination is promoted in coccidioidomycosis, however, they do not include works, which can provide more information in this regard and it would be very useful to include them in this section (for example Rodriguez-Ramirez et al., Mycopathologia 2018, 183(4):709-716; Kim et al., Can Assoc Radiol J 1998, 49(6):401-7).

4) I consider that section “4.1. Coccidioidomycosis in dogs”, is irrelevant to this work since it does not provide information on the main topic of this manuscript. In the mention made in reference 58 (Shubitz et al., 2021), the role of granulomas in dogs is not clear since, in this section only the finding of small granulomas in some of the dogs is mentioned located in the place where the injection of the vaccine tested in this study was applied. I suggest explaining what would be the appropriate immune response against Coccidioides? and postulating the protection mechanism during reexposure.

5) In the section “4.2. Valley fever in the lungs of mice and humans”, I suggest changing the title to “Valley fever in mice and humans.” In lines 333-334, the authors mention, “Spherules might transition upon macrophage engulfment of arthroconidia, switching into its parasitic life cycle”, they should correct this statement that is not clear; the transition is from arthroconidia to spherules with endospores, which is the parasitic phase.

During the process of transition from arthroconidia to spherules in the host macrophages, when the endospores are released, at what point could the granuloma form?

References

References should be checked carefully; for example, not all scientific names are italicized, some journals are abbreviated, and others are not.

Suggested references

Heidi G Rodriguez-Ramirez, Adolfo Soto-Dominguez, Gloria M González, Oralia Barboza-Quintana,Mario C Salinas-Carmona, Luis A Ceceñas-Falcon, Roberto Montes-de-Oca-Luna, Alma Y Arce-Mendoza, Adrian G Rosas-Taraco. Inflammatory and Anti-inflammatory Responses Co-exist Inside Lung Granuloma of Fatal Cases of Coccidioidomycosis: A Pilot Report. Mycopathologia 2018, 183(4):709-716. doi:10.1007/s11046-018-0264-7.

Kim KI, Leung AN, Flint JD, Müller NL. Chronic pulmonary coccidioidomycosis: computed tomographic and pathologic findings in 18 patients. Can Assoc Radiol J. 1998, 49(6):401-407.

Author Response

Response to Reviewer 2 Comments

The title of this paper suggests it is a review of granuloma formation in coccidioidomycosis and other infections. However, the use of vague and imprecise terms prevented me from understanding what histopathology the authors were describing. Even the word “granuloma” is imprecise, meaning simply a focal collection of inflammatory cells. The term is used by pathologists to describe such a wide variety of inflammatory responses that I don’t find the word descriptive. The authors of this paper also use the term, necrotizing granuloma (e,g, line 87). This could be a pyogranuloma with central neutrophils, a caseous granuloma with an amorphous noncellular center, or a mass of dead cells with surrounding inflammation.

In the context of Coccidioides, we do not know different types of granulomas. It would be wonderful to have a detailed way to describe these lung nodule formations during Coccidioides infections but with a limited number of studies in the field details are limited. If this reviwer is aware of other papers that describe Coccidioides granulomas in more detail, we may have missed, we would like to include them in this review as it would be valuable to the field. In Valley fever, clinicians typically use the plain term granuloma or necrotising granuloma to descibe these structures. As suggested by other reviewers, we have added details regarding CT imaging on granulomas (section 3.4). However, CT imaging does not identify granuloma types nor the cell types found within/surrounding the granulomas. We have utilized immunohistochemistry studies to highlight cell types in granulomas during infections. A couple of dog Coccidioides articles and a new chronic mouse model in Coccidioides have demonstrated pyogranuloma in these mammals. We have incorporated this term into the review within that context (lines 272 and 282).

Examples of other imprecise terms are “late appearing granulomas” (line 158),  “innate cells” (line232), “phagocytozing assay” (line 238), and “spore” lines 321-321, presumably meaning endospore or spherule.

We have modified to remove the impercise terms.

The section on CGD has a modest number of factual errors, such as stem cell transplantation not being effective in adults (line 254), usefulness of surgery in removing granulomas (line 239), defective phagocytosis (lines 231-232),  dehydrorhodamine measuring phagocytosis (line237), presence of sterile granulomas (line 235), etc.

We have edited this section to focus on CGD in the lung as this is the relavent tissue for Coccidioides infection. We have highlighted other reveiwes in this section that study the full CGD disease manifistaion for the readers’ information.

The discussion of coccidioidomycosis doesn’t include the well-described, neutrophil accumulation around ruptured spherules.

Thank you for suggesting to spend more time on neutrophils. In this section, we have added details on neutrophils (beginning of section 3.5 on line 262). Xue et al. shows neutrophils in mouse granulomas, but the data is not quantified and not from the lungs. If there are specific articles that you wish us to include, please let us know.

If the authors are correct that mature spherules are too large for ingestion by phagocytes, how do they explain failure of phagocytosis of the tiny endospores?

That is a good question that the field has not yet addressed. Macrophages have been more extensivly studied as they are easier to culture than neutrophils. Alhtough, there are some neutrophil fungal studies nd this area of research is expanding. We hope this question will be clarified based on those studies. Most studies macrohages studies focus on arthoconidia engulphment and trasnition into the spheurle stage, rather than spherule rupture and immune response to endospores. As we better understand the antigens at each stage of the life cycle, the immune response to each of these should be clear, but unfortunately this is currently unknown.

The discussion of inflammatory cell types in coccidioidomycosis (lines 336-375) seems a little out of place in a review of granuloma formation.

Macorphage function is critical to granuloma formation and maintenance in response to Coccidioides. The major cytokines produced by macrophages are pro- and anti-inflammatory thus, we felt it important to describe these general inflammatory responses in the review. These paragraphs also describe the figures and possible mechanisms involved in phagocytosis of Coccidioides.

Reviewer 3 Report

The title of this paper suggests it is a review of granuloma formation in coccidioidomycosis and other infections. However, the use of vague and imprecise terms prevented me from understanding what histopathology the authors were describing. Even the word “granuloma” is imprecise, meaning simply a focal collection of inflammatory cells. The term is used by pathologists to describe such a wide variety of inflammatory responses that I don’t find the word descriptive. The authors of this paper also use the term, necrotizing granuloma (e,g, line 87). This could be a pyogranuloma with central neutrophils, a caseous granuloma with an amorphous noncellular center, or a mass of dead cells with surrounding inflammation. Examples of other imprecise terms are “late appearing granulomas” (line 158), “innate cells” (line232), “phagocytozing assay” (line 238), and “spore” lines 321-321, presumably meaning endospore or spherule.

The section on CGD has a modest number of factual errors, such as stem cell transplantation not being effective in adults (line 254),  usefulness of surgery in removing granulomas (line 239), defective phagocytosis (lines 231-232),  dehydrorhodamine measuring phagocytosis (line237), presence of sterile granulomas (line 235), etc.

The discussion of coccidioidomycosis doesn’t include the well-described, neutrophil accumulation around ruptured spherules. If the authors are correct that mature spherules are too large for ingestion by phagocytes, how do they explain failure of phagocytosis of the tiny endospores?

The discussion of inflammatory cell types in coccidioidomycosis (lines 336-375) seems a little out of place in a review of granuloma formation.

Author Response

Response to Reviewer 3 Comments

The manuscript "Valley fever and other infectious granulomas: Advancements, gaps, and challenges" describes the current knowledge of granulomas in a general context, and particularly in Valley fever. A topic of great relevance, fundamentally for the medical area; however, I have several comments:

1) I suggest changing the title since the way it is written suggests that the central approach is granulomas in Valley fever; however, although I know that the information is very scarce on granulomas-Valley fever, most of the manuscript is focused on granulomas associated with other pathologies, which is where this clinical aspect is described in greater detail.

I suggest the title "Hypothesis on the presence of granulomas in Valley fever concerning other infectious and spontaneous diseases: Advances, gaps, and challenges."

Thank you for the suggestion, we decided to rename the title as ‘Coccidioidomycosis granulomas informed by other diseases: Advancements, gaps, and challenges.”

2) In section 3 “Granulomatous diseases,” it should be included as a subsection “3.4 Coccidioidomycosis.”

We have edited the sections and subsections.

3) In the section “4. Coccidiomycosis/Valley fever/Coccidioides” lines 280-282, the authors mention that more research on granuloma is needed to define essential cell types and phenotypes, to understand how chronic infection or dissemination is promoted in coccidioidomycosis, however, they do not include works, which can provide more information in this regard and it would be very useful to include them in this section (for example Rodriguez-Ramirez et al., Mycopathologia 2018, 183(4):709-716; Kim et al., Can Assoc Radiol J 1998, 49(6):401-7).

This wounderful paper (Rodriguez-Ramirez et al.,) is already cited in this review (citation 33). It is now included in section 3.5. As for Kim et al., we have been unable access this full article (even through mulitple attemps via UC Interlibrary loan). On PubMed, the full article is not accessesable. In place of the Kim et. al. manuscript, we have added Capone et al., (citation 98) to discuss acute Coccidioides.

4) I consider that section “4.1. Coccidioidomycosis in dogs”, is irrelevant to this work since it does not provide information on the main topic of this manuscript. In the mention made in reference 58 (Shubitz et al., 2021), the role of granulomas in dogs is not clear since, in this section only the finding of small granulomas in some of the dogs is mentioned located in the place where the injection of the vaccine tested in this study was applied. I suggest explaining what would be the appropriate immune response against Coccidioides? and postulating the protection mechanism during reexposure.

This dog section has been deleted and the vaccine components clarified.

5) In the section “4.2. Valley fever in the lungs of mice and humans”, I suggest changing the title to “Valley fever in mice and humans.”

We have modified the section tittle.

In lines 333-334, the authors mention, “Spherules might transition upon macrophage engulfment of arthroconidia, switching into its parasitic life cycle”, they should correct this statement that is not clear; the transition is from arthroconidia to spherules with endospores, which is the parasitic phase.

This sentence has been fixed.

During the process of transition from arthroconidia to spherules in the host macrophages, when the endospores are released, at what point could the granuloma form?

That is a fantatsic question for which the field does not have an answer. The best information about the timing of granumola formation is found the recent non-human primate TB study (reference 21).

References

References should be checked carefully; for example, not all scientific names are italicized, some journals are abbreviated, and others are not.

Thank you for ctaching this, our citation manager did not inport all of the citations correctly. We believe this has been corrected and will work with the journal editor as needed if other issues are present.

Suggested references

Heidi G Rodriguez-Ramirez, Adolfo Soto-Dominguez, Gloria M González, Oralia Barboza-Quintana,Mario C Salinas-Carmona, Luis A Ceceñas-Falcon, Roberto Montes-de-Oca-Luna, Alma Y Arce-Mendoza, Adrian G Rosas-Taraco. Inflammatory and Anti-inflammatory Responses Co-exist Inside Lung Granuloma of Fatal Cases of Coccidioidomycosis: A Pilot Report. Mycopathologia 2018, 183(4):709-716. doi:10.1007/s11046-018-0264-7.

Citation already in publication it was one of the messed up citation 33.

Kim KI, Leung AN, Flint JD, Müller NL. Chronic pulmonary coccidioidomycosis: computed tomographic and pathologic findings in 18 patients. Can Assoc Radiol J. 1998, 49(6):401-407.

Curruently unable to access the full manuscript. Instead used:

Capone D, Marchiori E, Wanke B, Dantas KE, Cavalcanti MA, Deus Filho A, Escuissato DL, Warszawiak D. Acute pulmonary coccidioidomycosis: CT findings from 15 patients. Br J Radiol. 2008 Sep;81(969):721-4. doi: 10.1259/bjr/12054884. Epub 2008 May 28. PMID: 18508875.

Round 2

Reviewer 3 Report

The revised manuscript is adequate.